# Detection of *Streptococcus pyogenes* M1_{UK} in Australia and characterization of the mutation driving enhanced expression of superantigen SpeA

Mark R. Davies [1,13] ✉, Nadia Keller[2,13], Stephan Brouwer [2,13], Magnus G. Jespersen [1,13], Amanda J. Cork [2], Andrew J. Hayes [1], Miranda E. Pitt [1], David M. P. De Oliveira [2], Nichaela Harbison-Price [2], Olivia M. Bertolla[2], Daniel G. Mediati[3], Bodie F. Curren [2], George Taiaroa[1], Jake A. Lacey [4], Helen V. Smith [5], Ning-Xia Fang[5], Lachlan J. M. Coin[1], Kerrie Stevens[6], Steven Y. C. Tong [4,7], Martina Sanderson-Smith [8], Jai J. Tree[3], Adam D. Irwin [9,10], Keith Grimwood[11,12], Benjamin P. Howden [6], Amy V. Jennison[5] & Mark J. Walker [2] ✉

A new variant of *Streptococcus pyogenes* serotype M1 (designated 'M1_{UK}') has been reported in the United Kingdom, linked with seasonal scarlet fever surges, marked increase in invasive infections, and exhibiting enhanced expression of the superantigen SpeA. The progenitor *S. pyogenes* 'M1_{global}' and M1_{UK} clones can be differentiated by 27 SNPs and 4 indels, yet the mechanism for *speA* upregulation is unknown. Here we investigate the previously unappreciated expansion of M1_{UK} in Australia, now isolated from the majority of serious infections caused by serotype M1 *S. pyogenes*. M1_{UK} sub-lineages circulating in Australia also contain a novel toxin repertoire associated with epidemic scarlet fever causing *S. pyogenes* in Asia. A single SNP in the 5' transcriptional leader sequence of the transfer-messenger RNA gene *ssrA* drives enhanced SpeA superantigen expression as a result of *ssrA* terminator readthrough in the M1_{UK} lineage. This represents a previously unappreciated mechanism of toxin expression and urges enhanced international surveillance.

*Streptococcus pyogenes* (commonly referred to as the group A *Streptococcus*) is a strictly human pathogen of global health significance, accounting for over 500,000 deaths worldwide per year[1–3]. *S. pyogenes* also causes scarlet fever, occurring primarily in children aged 5–15 years[1,3]. Defining symptoms include a confluent, deep red, sandpaper-like rash, "strawberry tongue", and exudative tonsillopharyngitis. While a major cause of childhood morbidity with 15–20% infection mortality rate in the 19th and early 20th centuries, scarlet fever had been in decline as a public health threat for over 100 years[1,4]. The re-emergence of scarlet fever in the United Kingdom (UK), Hong Kong

and mainland China[5–8] is a new public health threat. Asian scarlet fever outbreak isolates carry mobile genetic elements encoding antibiotic resistance (tetracycline, erythromycin and clindamycin) and highly potent toxins, including the superantigens SSA and SpeC, and the DNase Spd1[6,8–10].

*S. pyogenes* strains are classified into over 250 *emm*-types by sequencing the 5' end of the gene encoding the serotype-defining M protein (*emm*)[11,12]. In China and Hong Kong, the most common *emm*-types causing scarlet fever are *emm*12 and *emm*1[6,8,13]. UK *emm*-types commonly associated with scarlet fever are *emm*1, *emm*12, *emm*3 and

*emm*4 *S. pyogenes*[7,14]. Serotype M1 *S. pyogenes* (*emm*1; the 'M1T1 clone', here designated 'M1$_{global}$'), has been the major driver of invasive infections in Western countries since the mid-1980s[1,15–18]. Reports in 2019 from the UK describe the rapid emergence of a new *S. pyogenes emm*1 clonal lineage (M1$_{UK}$) contributing to seasonal surges in scarlet fever and a marked increase in invasive infections, exhibiting enhanced expression of the superantigen SpeA (a key virulence factor of *S. pyogenes*). M1$_{UK}$ is differentiated from M1$_{global}$ by 27 chromosomal single nucleotide polymorphisms (SNPs)[19,20].

Here, we demonstrate the unappreciated expansion of the M1$_{UK}$ lineage in Australia, with sub-lineages containing a novel toxin gene repertoire of *ssa*, *speC* and *spd*1. We provide new mechanistic insight

into *S. pyogenes* toxin regulation by demonstrating that a SNP in the 5' transcriptional leader of the transfer-messenger RNA (tmRNA - encoded by the *ssrA* gene) drives increased SpeA superantigen expression in the M1$_{UK}$ lineage through transcriptional *ssrA* terminator read-through into the *speA* operon reading frame.

## Results

### Detection of *Streptococcus pyogenes* M1$_{UK}$ in Australia

The emergence of the M1$_{UK}$ lineage in the UK and its detection in other countries[19,21–23] triggered our investigation of 318 Australian *emm*1 *S. pyogenes* isolates. Overall 310/318 were invasive isolates from sterile body sites and sourced from state-based public health laboratories in

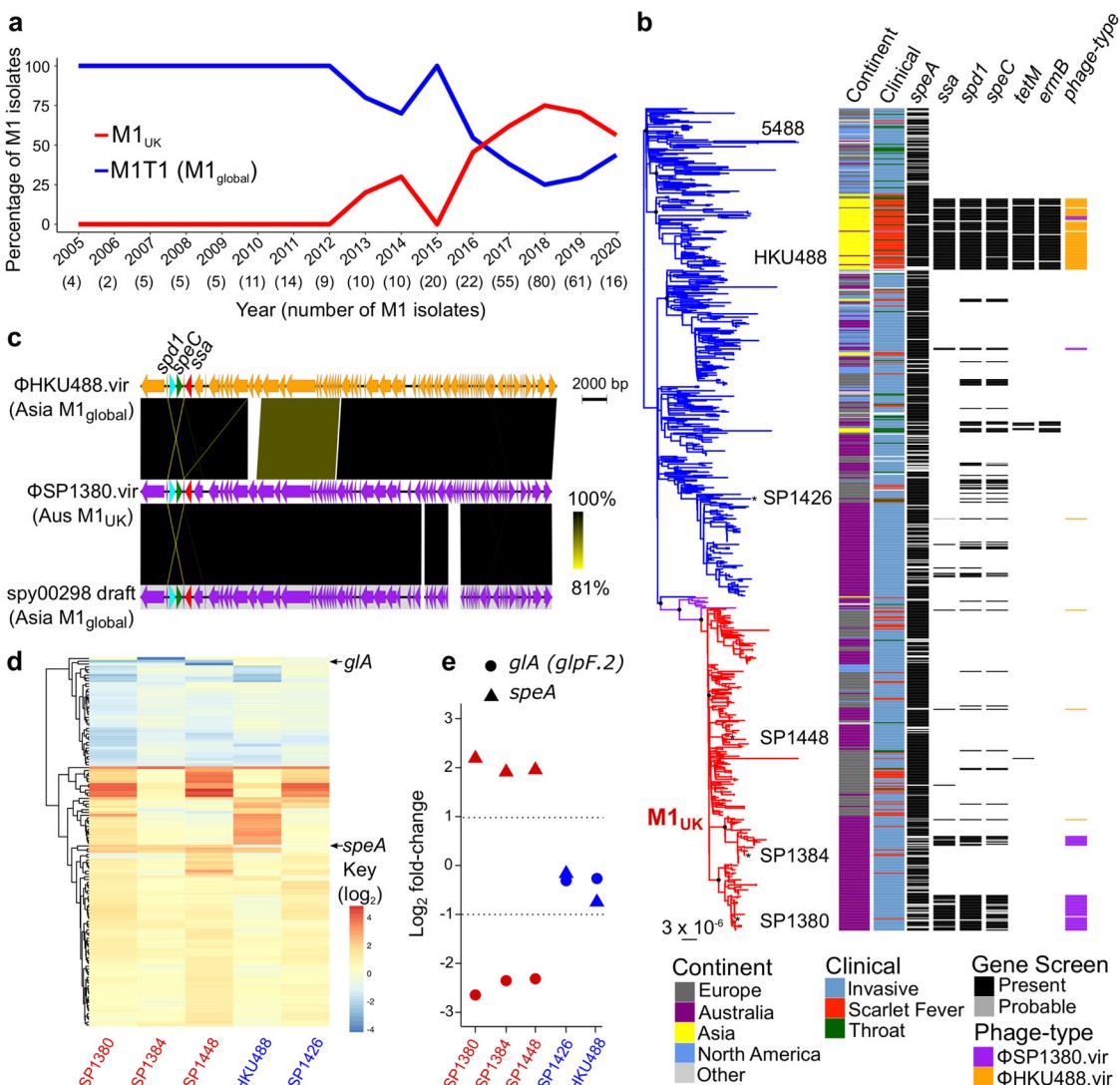

**Fig. 1 | Characterization of M1$_{UK}$ genotype in Australia. a** Frequency of two M1 genotypes; M1$_{UK}$ (red) and M1$_{global}$ (blue) in representative clinical specimens from Queensland and Victoria between 2005 and 2019. Definition of M1$_{UK}$ is based on the presence of 27 defining SNPs and 4 indels (Supplementary Table 1). **b** Maximum-likelihood phylogenetic tree of 737 Australian and global M1 isolates built on 3465 SNP sites from a 1,623,078 bp core genome alignment relative to the M1$_{global}$ 5448 reference genome. Black circles at major branch nodes refer to >90% bootstrap support. Branches are coloured according to genetic sub-populations; ancestral M1$_{global}$ (blue), intermediate SNP profile M1$_{inter}$ (purple), and 27 SNP and 4 deletions M1$_{UK}$ (red). Selected type strains are annotated. Locality and clinical sample type are coloured as per legend provided. Carriage of bacteriophage-encoded toxin genes and antibiotic resistance determinants are indicated by black blocks, with contig fragmented hits indicated by grey blocks. **c** Pairwise tblastN comparison of

*speC*, *ssa* and *spd*1 carrying prophage ΦHKU488.vir, ΦSP1380.vir and a draft prophage genome spy00298 representing overall high sequence similarity with prophage colour coded as per distribution in (**b**). Sequence diversity is scaled from 100% (black) to 80% (yellow). **d** Heatmap of significantly differential expressed genes (*p* > 0.05, ≥2 fold change; *n* = 3) in Australian M1$_{UK}$ genotype strains (SP1380, SP1384, SP1448 in red) and M1$_{global}$ strains (SP1426, HKU488 in blue) relative to the M1$_{global}$ reference strain 5448. Key represents log$_2$ fold-change (refer to Supplementary Table 4 for values). Histogram of related gene expression profiles is shown on the edge of the heatmap. **e** Differential expressed data as per (**d**) where only genes commonly differentially expressed between M1$_{UK}$ genotype strains (SP1380, SP1384, SP1448) and M1$_{global}$ strains SP1426, HKU488 are shown. Source data are provided as a Source Data file.

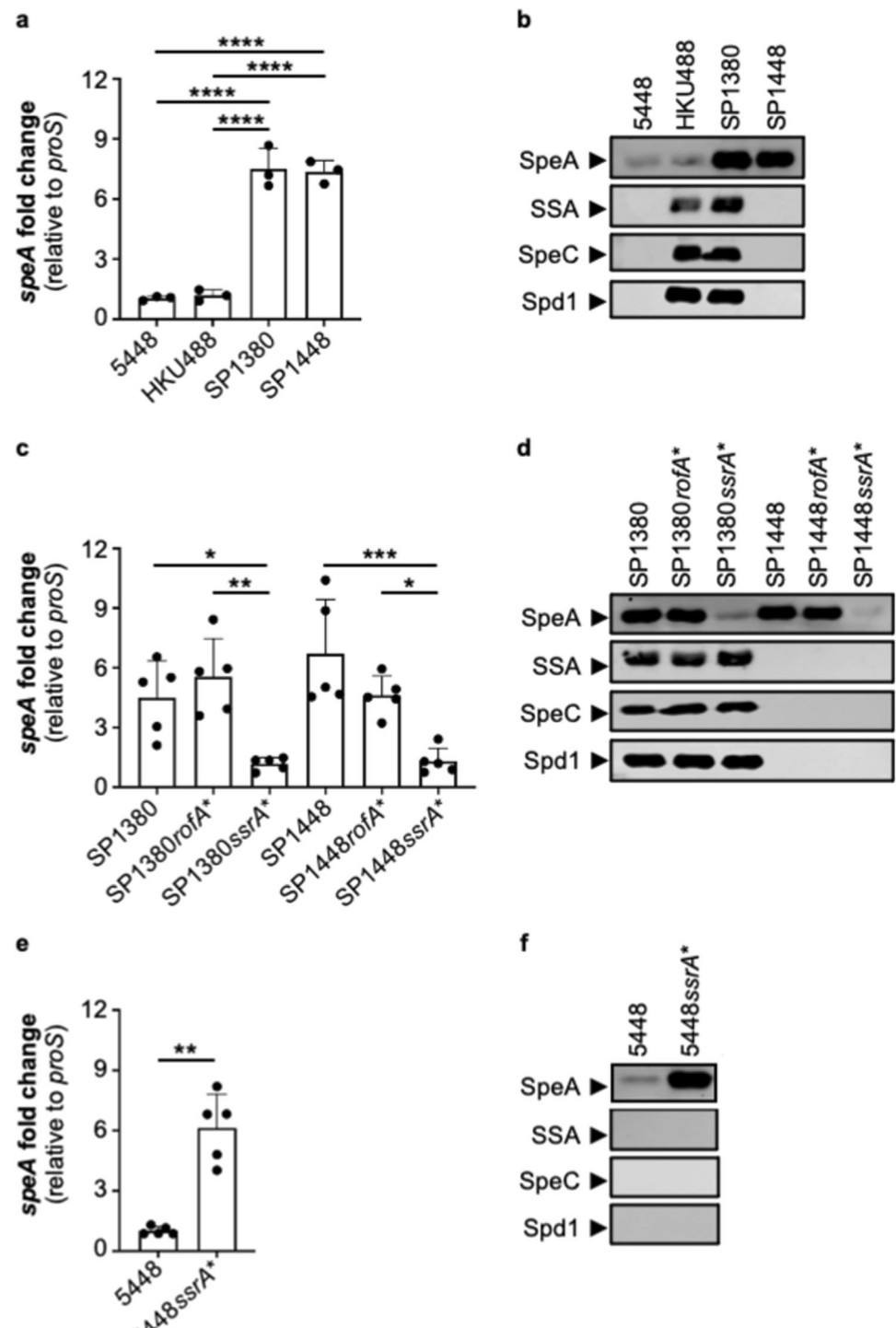

**Fig. 2 | A single +5 G > C SNP in the 5' leader sequence of the small noncoding RNA *ssrA* is responsible for increased SpeA expression in M1$_{UK}$.**
**a, c, e** Quantitative real-time PCR determining *speA* mRNA expression levels in 5448 (M1$_{global}$), HKU488 (M1$_{global}$), SP1380 (M1$_{UK}$), SP1448 (M1$_{UK}$), SP1380$^{rofA*}$ (three *rofA* SNPs repaired), SP1380$^{ssrA*}$ (single *ssrA* SNP repaired), SP1448$^{rofA*}$ (three *rofA* SNPs repaired), SP1448$^{ssrA*}$ (single *ssrA* SNP repaired) and 5448$^{ssrA*}$ (single *ssrA* SNP introduced). Data from at least three biological replicates are presented as mean

values ± SD (**a** $n = 3$, **c** $n = 5$, **e** $n = 5$). Statistical significance was assessed using one-way ANOVA with Tukey's multiple comparisons post hoc test (**a** ****$p < 0.0001$; **c** SP1380 vs. SP1380$^{ssrA*}$ *$p = 0.0395$, SP1380$^{rofA*}$ vs. SP1380$^{ssrA*}$ **$p = 0.0037$, SP1448 vs. SP1448$^{ssrA*}$ ***$p = 0.0003$, SP1448$^{rofA*}$ vs. SP1448$^{ssrA*}$ *$p = 0.0403$) and Welch's t test (**e** 5448$^{ssrA*}$ **$p = 0.0023$). **b, d, f** Western immunoblot detection of bacteriophage-encoded superantigens SpeA, SSA, and SpeC and DNase Spd1 in culture supernatants ($n = 1$). Source data are provided as a Source Data file.

Queensland and Victoria between 2005 and 2020. The remaining 8 isolates were from the throats of children diagnosed with scarlet fever. In addition to the defining 27 M1$_{UK}$ SNPs[19], we also defined 4 small deletion events (3 single base pair intergenic deletions and one in-frame 3 bp deletion) that were omnipresent in the M1$_{UK}$ genotype

analyzed in this study (Supplementary Table 1). Plotting the frequency of the *emm*1 genotype since 2005 revealed the rapid expansion of M1$_{UK}$ in Australia, with >60% of clinical *emm*1 *S. pyogenes* being of the M1$_{UK}$ genotype by 2019 (Fig. 1a). Phylogenetic comparison of 737 *emm*1 *S. pyogenes* genomes from Europe, North America, Asia and

Australia supports the proposal of a single common ancestor for the progenitor M1$_{UK}$ population[19] irrespective of geographical source, indicative of pandemic spread (Fig. 1b). Analysis of the accessory genome content of the emm1 population found that 26% of all Australian M1$_{UK}$ strains have subsequently acquired the ssa, speC, and spd1 toxin repertoire (Fig. 1b) which is also over-represented in Asian M1$_{global}$ strains[6,9,24].

To examine whether the Australian M1$_{UK}$ strains harbour a related prophage to Asian M1$_{global}$ strains, the complete genome of an Australian M1$_{UK}$ strain SP1380 carrying the ssa, speC, and spd1 toxin repertoire was determined. The 1,883,075 bp SP1380 genome exhibited typical M1 genome features such as three prophage regions - ΦSP1380.1 carrying speA (chromosomal site H); ΦSP1380.2 carrying spd3 (chromosomal site K) and ΦSP1380.3 carrying sdaD2 (also designated Sda1; chromosomal site O) in addition to a fourth prophage region carrying ssa, speC and spd1, termed ΦSP1380.vir (chromosomal site Q) (Supplementary Fig. 1). ΦSP1380.vir has 95% similarity to the Hong Kong scarlet fever outbreak prophage ΦHKU488.vir (Fig. 1c). Comparative analysis of ΦSP1380.vir with the broader emm1 phage population revealed the presence of ΦSP1380.vir in 5 M1$_{global}$ strains (Fig. 1b, c), indicative of probable prophage convergence within Australian M1$_{global}$ and M1$_{UK}$ genotypes.

### A single SNP in the 5′ transcriptional leader sequence of the tmRNA gene ssrA drives enhanced M1$_{UK}$ SpeA superantigen production

To investigate the impact of the 27 M1$_{UK}$ lineage-defining SNPs and 4 deletions on global gene transcription, we performed complete genome sequencing and RNA-seq analysis of three Australian M1$_{UK}$ genotype S. pyogenes isolates; SP1380 (scarlet fever; ssa$^+$, speC$^+$, spd1$^+$, speA$^+$), SP1384 (scarlet fever; ssa$^-$, speC$^-$, spd1$^-$, speA$^+$), and SP1448 (invasive disease; ssa$^-$, speC$^-$, spd1$^-$, speA$^+$) (Fig. 1b and Supplementary Fig. 2). SP1380, SP1384, and SP1448 contain the M1$_{UK}$ lineage-defining 27 SNPs and 4 deletions (Supplementary Table 1). S. pyogenes M1$_{global}$ genotype strains 5448 (invasive disease; ssa$^-$, speC$^-$, spd1$^-$, speA$^+$)[25], HKU488 (scarlet fever; ssa$^+$, speC$^+$, spd1$^+$, speA$^+$)[24] and an Australian S. pyogenes M1$_{global}$ clinical isolate SP1426 (scarlet fever; ssa$^-$, speC$^+$, spd1$^+$, speA$^+$) were used as benchmark reference strains for comparison with Australian M1$_{UK}$ strains (Fig. 1b and Supplementary Fig. 2). While small levels of transcriptional heterogeneity exist across M1$_{UK}$ strains when mapped to M1$_{global}$ 5448 (Fig. 1d), RNA-seq analysis revealed that only two genes were commonly differentially regulated in the 3 M1$_{UK}$ genotype strains compared to the 3 strains representing the M1$_{global}$ clone (Fig. 1e). As expected, speA was upregulated while the gene encoding for a putative glycerol facilitator aquaporin glA (glpF.2) was significantly downregulated, likely as a direct result of a M1$_{UK}$ lineage-defining SNP located in the promoter region of the glA gene (Supplementary Table 4). Validating these and published findings from the UK[19], qPCR and western blot analysis of SpeA in M1$_{UK}$ strains SP1380, SP1384, and SP1448 showed a ~5-fold increase in speA gene transcripts and significantly higher levels of SpeA in culture supernatants in comparison to the M1$_{global}$ strains SP1426, 5448 and HKU488 (Fig. 2a, b). As expected, both HKU488 and SP1380 expressed the full repertoire of scarlet fever-associated superantigens SSA and SpeC, and the DNase Spd1 (Fig. 2b).

To identify which M1$_{UK}$ lineage-defining genetic features (Supplementary Table 1) result in upregulation of speA expression, we constructed sets of isogenic mutants using M1$_{UK}$ strains SP1380 and SP1448, and the S. pyogenes M1$_{global}$ reference strain 5448. The S. pyogenes virulence regulators RofA and Nra[26,27] have been implicated in speA gene regulation in M6 and M49 S. pyogenes[27,28] with three missense rofA SNPs plausibly postulated to cause increased speA superantigen expression in the M1$_{UK}$ lineage[19]. To test this hypothesis, we firstly constructed isogenic mutants in the wildtype SP1380 and SP1448 genetic backgrounds, with the 3 rofA SNPs corrected to reflect the M1$_{global}$ genotype (SP1380$^{rofA*}$, SP1448$^{rofA*}$). SpeA expression was unaffected by the repair of the 3 rofA SNPs (Figs. 2c and 2d) and no other differentially expressed genes were observed across the genome as assessed by RNA-seq under the conditions tested (Supplementary Fig. 3). Next, we chose to investigate the SNP (+5 G > C) found in the 26 nucleotide 5′-leader sequence of tmRNA encoded by the ssrA gene[29–31], located ~1 kb upstream of the speA gene and adjacent to the predicted bacterial attachment site (attB) into which speA-encoding prophages integrate into the M1$_{global}$ genome[32]. The ssrA gene encodes a component of the conserved bacterial ribosome rescue system with dual alanine-tRNA-like and mRNA-like properties[33,34]. Correction of the single 5′ transcriptional leader ssrA SNP in the SP1380 and SP1448 M1$_{UK}$ genetic backgrounds, to reflect the progenitor M1$_{global}$-like genotype (SP1380$^{ssrA*}$, SP1448$^{ssrA*}$), resulted in a significant reduction in transcripts and protein expression of SpeA (Fig. 2c, d; Supplementary Fig. 3). To validate this finding, we introduced the M1$_{UK}$ 5′ transcriptional leader ssrA SNP into the 5448 M1$_{global}$ genetic background (5448$^{ssrA*}$) which resulted in a ~5-fold increase in speA transcripts (Fig. 2e; Supplementary Fig. 3). This increase is equivalent to levels detected in the Australian SP1380, SP1384 and SP1448 M1$_{UK}$ strains (Fig. 2a). As predicted, SpeA protein levels were also markedly increased in 5448$^{ssrA*}$ (Fig. 2f). An additional 5 genes encompassing a putative membrane transport protein and genes within the carbohydrate utilization Lac.2 operon[35] were also differentially expressed in 5448$^{ssrA*}$ compared to wildtype 5448 (Supplementary Table 1). The prophage associated paratox (ptx) gene which is located between ssrA and speA in modern emm1 genotypes was not differentially transcribed in M1$_{UK}$ compared to M1$_{global}$, or in the 5′ transcriptional leader ssrA isogenic mutant set. This finding was to be expected considering that the paratox open reading frame is predicted to be transcribed from the anti-sense strand. These loss- and gain-of-function studies demonstrate that the single 5′ transcriptional leader ssrA SNP represents a critical molecular event that is necessary and sufficient for increased SpeA production in the M1$_{UK}$ lineage.

### The M1$_{UK}$ 5′ transcriptional leader ssrA gene SNP drives enhanced SpeA superantigen expression as a result of ssrA terminator read-through

Little is known about transcriptional control of the speA gene in emm1 S. pyogenes and no transcriptional regulator for the putative speA promoter has been identified[36]. SpeA expression can be detected in all phases of growth in vitro and is found to peak in late logarithmic growth phase[37]. Considering these data, our finding that the 5′ transcriptional leader ssrA SNP alters SpeA production was unexpected. To investigate how the SNP in the 5′ leader of ssrA affects downstream speA transcription, we analyzed the local read coverage around the speA-phage integration site using the SP1380 isogenic strain set (Fig. 3a). RNA-seq data suggest that 0.25−0.35% of ssrA transcripts read past a predicted ssrA terminator[38] through into the speA gene of the Australian M1$_{UK}$ SP1380 (Fig. 3a, Supplementary Fig. 4). This level of ssrA transcriptional read-through was equivalent in the SP1380$^{rofA*}$ background, yet 5 times reduced (0.05−0.08%) in SP1380$^{ssrA*}$ (Fig. 3a and Supplementary Fig. 4). This change in ssrA transcriptional read-through was similar to the increase in speA gene transcripts detected by qPCR (Fig. 2c). Notably, the transcriptional profile of SP1380$^{ssrA*}$ resembled that of the M1$_{global}$ genotype 5448 whereas 5448$^{ssrA*}$ showed enhanced ssrA transcriptional read-through (0.23−0.26%), underscoring the critical role of the 5′ transcriptional leader ssrA SNP in enhanced speA expression (Supplementary Fig. 4). Transcription of ssrA itself remained unchanged in all strains analyzed, indicating that the 5′ transcriptional leader ssrA SNP does not alter ssrA promoter activity in M1$_{UK}$, compared to M1$_{global}$ (Fig. 3a).

To validate the preliminary findings that transcriptional read-through from ssrA is evident, we undertook native RNA sequencing of the SP1380 strain using the long-read Oxford Nanopore Technologies

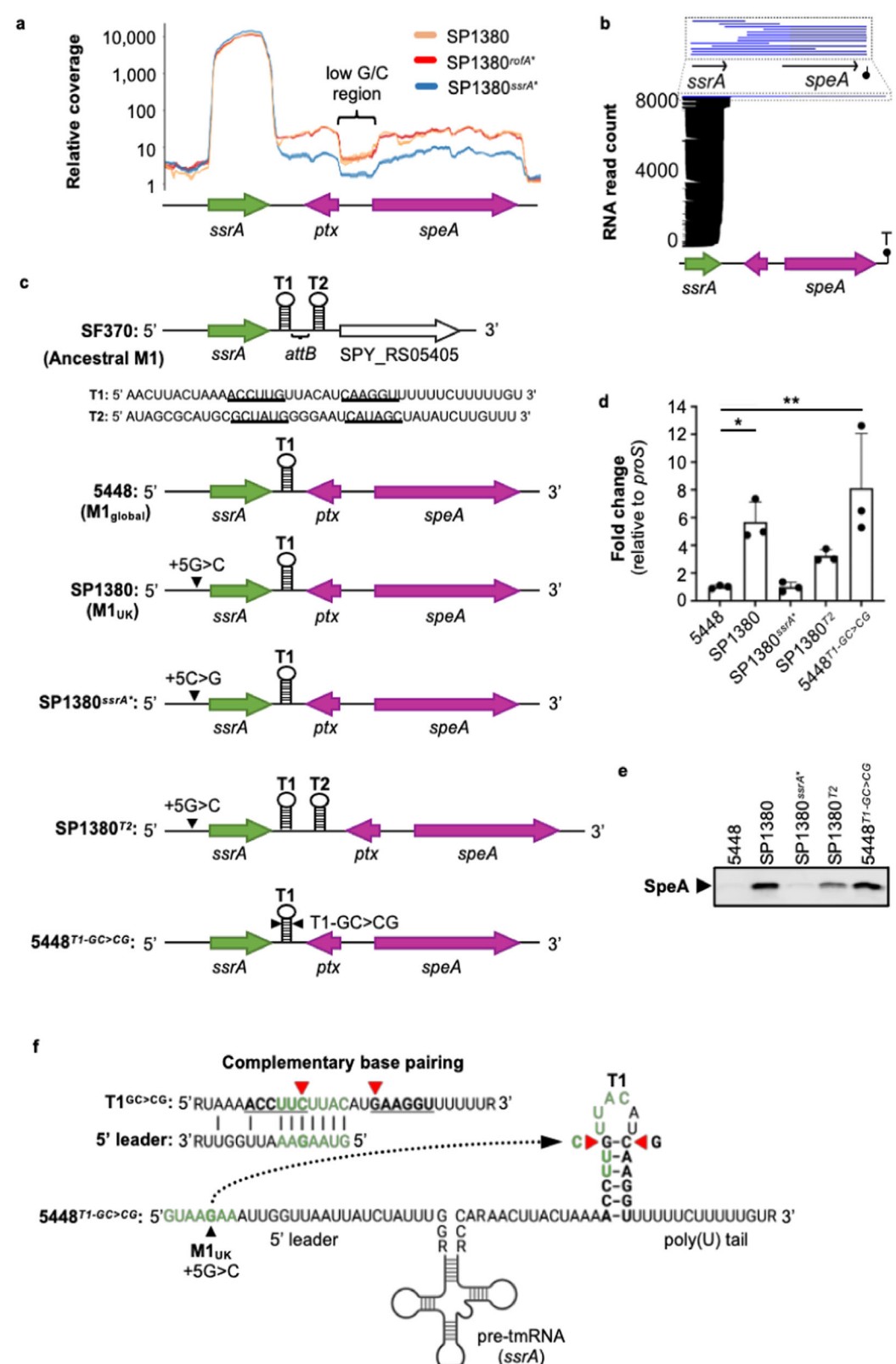

(ONT) platform[39–41]. Plotting of native RNA reads to the SP1380 *ssrA* and *speA* genomic region revealed the presence of single RNA transcripts that originated within *ssrA* and extended through into the *speA* open reading frame (Fig. 3b). Several RNA reads ranging from 1692 to 1840 bp in size extended from *ssrA* through to a predicted *speA* terminator (as defined by ARNold[42], SP1380 genome coordinates 1,008,326 to 1,008,346 bp). Degradation of RNA transcripts was

evident yet is not unexpected given the nature of long-read native RNA sample processing and sequencing. Consistent with the RNA sequencing results, Northern blot analysis probing for *speA* in SP1380 verified a ~1.8 kb transcript that correlates with the predicted size of the *ssrA*-*speA* bicistronic RNA (Supplementary Fig. 5). Of note, an additional 0.9 kb *speA* fragment was evident in the SP1380 Northern blot that increased with the *ssrA*-*speA* transcript, suggesting that a

**Fig. 3 | High-level *speA* expression in M1$_{UK}$ results from increased transcriptional read-through of the *ssrA* gene. a** Enrichment of RNA-seq reads on the positive strand showing increased transcriptional read-through from *ssrA* into *speA* in M1$_{UK}$ (SP1380) which is dependent on the single *ssrA* SNP (SP1380$^{ssrA*}$), but not the three *rofA* SNPs (SP1380$^{rofA*}$). The indicated low G/C region results in reduced mapping coverage to this genomic region. The paratox gene (*ptx*) which typically flanks phage toxin genes in *S. pyogenes* is located on the negative strand. **b** Read stack of native RNA transcripts within SP1380 (M1$_{UK}$) *ssrA* to *speA* regulon as determined by ONT long-read sequencing. *Y*-axis refers to the number of individual reads with each bar running along the horizontal referring to RNA read-position (and total length) relative to the *ssrA-speA* region (SP1380 genome coordinates 1,006,535 to 1,008,346 bp). Location of predicted *speA* transcriptional terminator ('T') is indicated (coordinates 1,008,326 to 1,008,346 bp). RNA reads extending beyond the predicted *ssrA* terminator (transcriptional read-through) are coloured blue. The inset above displays RNA transcripts that extend past the *ssrA* terminator. A total of 0.7% of the RNA reads spanned within *ssrA* into *speA* transcript. **c** Schematic representation of the *ssrA* to *speA* genomic region in M1 *S. pyogenes* genotypes. *ssrA* in ancestral M1 *S. pyogenes* (SF370) comprises two Rho-independent transcriptional terminators (T1 and T2). Palindromic sequences of

Rho-independent terminators that form a stem loop in T1 and T2 are underlined. *speA*-phage integration site in M1$_{global}$ genotypes (5448) occurs between T1 and T2 at the bacterial prophage attachment site (*attB*). The genomic location of the 5' *ssrA* SNP (+5 G > C) in M1$_{UK}$ is indicated by a black triangle. T2 was reintroduced into the M1$_{UK}$ background to study transcriptional read-through in SP1380$^{T2}$. Two point mutations were introduced into T1 of the M1$_{global}$ strain 5448 (5448$^{T1-GC>CG}$) to assess the effect of *ssrA* leader sequence and T1 stem structure base pairing on transcriptional read-through. **d** Quantitative real-time PCR of *speA* gene expression. Statistical significance was assessed using one-way ANOVA with Dunnett's multiple comparisons post-hoc test against the control strain 5448 (SP1380 *$p$ = 0.0398, 5448$^{T1-GC>CG}$ **$p$ = 0.0033; $n$ = 3). Data are presented as mean values ± SD. **e** Western immunoblot detection of SpeA protein abundance showing reduced expression in SP1380$^{T2}$ and increased expression in 5448$^{T1-GC>CG}$ ($n$ = 1). **f** Structure of the *ssrA* transcript showing secondary structure of the *ssrA* pre-tmRNA and T1 stem loop. Sequence complementary between the 5' transcriptional leader of *ssrA* and T1 terminator stem loop is highlighted (green). Position of red triangles indicates the GC > CG mutation in the T1 stem-loop (5448$^{T1-GC>CG}$). Source data are provided as a Source Data file.

monocistronic *speA* transcript is generated by processing of the bicistronic *ssrA-speA* transcript. Cleavage of the bicistronic transcript may occur during tmRNA maturation of the *ssrA* transcript that requires 3' end processing by endoribonucleases[43]. The abundance of the *ssrA-speA* transcript increased in SP1380 (M1$_{UK}$) and was restored to M1$_{global}$-like levels in 5448 and the SP1380$^{ssrA*}$ strain. These data support read-through transcription of *speA* from the upstream *ssrA* promoter leading to increased amounts of *ssrA-speA* transcripts in the M1$_{UK}$ genetic background.

These findings indicate that *ssrA* transcriptional read-through may drive *speA* expression in the M1$_{global}$ genetic background. To understand how φSP1380.1 phage insertion has coupled *ssrA* and *speA* transcription, we compared the *ssrA* genetic context of the ancestral M1 genotype (archetypical strain SF370) to the modern M1 genotype (M1$_{global}$ and M1$_{UK}$). In the ancestral (pre-1980s) SF370 genotype that lacks the *speA* prophage, two predicted Rho-independent terminators (T1 and T2) are present downstream of *ssrA* (Fig. 3c). In modern M1$_{global}$ and M1$_{UK}$ lineages, T2 is disrupted by *speA* prophage integration[32] (Fig. 3c). We hypothesized that partial 3' extension of the *ssrA* transcript occurs past the T1 terminator but transcription of *ssrA* is efficiently terminated at the T2 terminator in SF370 (Fig. 3c). Indeed, mapping of RNA-seq data in the SF370 background identified low levels of *ssrA* transcriptional read-through past the T1 terminator, yet effective termination at T2 (Supplementary Fig. 4). Furthermore, re-insertion of the *ssrA* T2 terminator sequence from SF370 into SP1380 (to the ancestral SF370-like form; SP1380$^{T2}$) partially reduced *speA* expression in the M1$_{UK}$ genetic background, compared to wildtype SP1380 (Fig. 3d). The reduction in SpeA production in SP1380$^{T2}$ was confirmed by western blot (Fig. 3e). Finally, we hypothesized that complementarity between the first 7 nucleotides of the *ssrA* leader and the T1 terminator stem loop sequence, which is enhanced by the M1$_{UK}$ 5' transcriptional leader *ssrA* +5 G > C SNP (Fig. 3f), results in T1 terminator unfolding and increased transcriptional read-through. To test this hypothesis, we constructed 5448$^{T1-GC>CG}$ by introducing two point mutations in the T1 sequence of M1$_{global}$ strain 5448. This change creates the same 7 nucleotides of complementarity with the 5448 *ssrA* leader sequence whilst retaining base pairing within the T1 terminator stem structure (Fig. 3c, f). Expression of SpeA was enhanced to levels equivalent to that of the M1$_{UK}$ strain SP1380 indicating that complementarity between the *ssrA* transcriptional leader sequence and T1 terminator promotes *ssrA* read-through and *speA* expression (Fig. 3d, e). Collectively, these data demonstrate that *speA* expression in the M1$_{global}$ and M1$_{UK}$ lineage is associated with transcriptional read-through from the *ssrA* promotor caused by *speA* prophage integration

between *ssrA* terminators, which is further enhanced in the M1$_{UK}$ sub-population by the +5 G > C SNP in the 5' leader sequence of *ssrA*.

## Discussion

The *S. pyogenes* M1$_{global}$ (M1T1) clone emerged in the 1980s, which paralleled an increase in severe invasive disease. The M1$_{global}$ clone subsequently disseminated worldwide, accounting for a significant proportion of clinical isolates within high-income settings[1,15–18]. Three horizontally acquired genetic events differentiate the M1$_{global}$ clone from other *emm*1 strains circulating at that time: homologous replacement of a 36 kb chromosomal region encoding the toxins NAD-glycohydrolase and streptolysin O and acquisition of two bacteriophages that encode the DNase SdaD2 (Sda1) and the superantigen SpeA[15–18]. The SpeA-encoding bacteriophage inserted into the *S. pyogenes* chromosome directly downstream of the *ssrA* gene[32]. The rapid emergence of the new M1$_{UK}$ variant as the dominant *emm*1 sub-clone in the UK[19,20] and Netherlands[21], and subsequent detection in North America[22,23], demands a thorough epidemiological assessment of the global public health threat that this new *S. pyogenes* variant poses. We reveal rapid replacement of the M1$_{global}$ genotype with M1$_{UK}$ in cases of severe infections identified in two populous Australian states. Furthermore, 26% of Australian M1$_{UK}$ strains have acquired the bacteriophage-encoded superantigens SSA and SpeC, and the DNase Spd1. This toxin repertoire is over-represented in Asian M1$_{global}$ and M12 isolates causing epidemic scarlet fever[6,8,9,24]. Bacteriophage-mediated horizontal transfer of bacterial virulence determinants may increase bacterial strain diversity and improve evolutionary fitness[10,44–47], driving the expansion of the M1$_{UK}$ lineage in the human population.

Scarlet fever isolates circulating in Asia are associated with a repertoire of toxin genes, which encode superantigens *ssa* and *speC*, and the DNase *spd*1 toxin[6,8–10,24]. In Australia, 26% of circulating M1$_{UK}$ sub-lineages also contain this novel toxin gene repertoire, suggesting independent acquisition of mobile genetic elements into distinct M1$_{UK}$ sub-lineages, likely as a result of strong positive selection pressure. The contribution of SSA, SpeC, and Spd1 to intranasal colonization of HLA-B6 mice has been explored in an *emm*12 scarlet fever isolate[10], and future studies to determine the contribution of SSA, SpeC, and Spd1 to M1$_{UK}$ virulence are warranted.

In bacteria, *ssrA* RNA (also known as tmRNA or 10Sa RNA) acts first as a tRNA to bind stalled ribosomes, then as an mRNA to tag the nascent polypeptides for degradation in a process termed ribosome rescue[33,34]. Bacterial *ssrA* is a hotspot for insertion of mobile genetic elements[48]. In *S. pyogenes*, *ssrA* is the insertion site of multiple phage carrying *speA* and other toxins[49] which in M1 *S. pyogenes* occurs

between two Rho-independent terminators, affecting efficient termination of the *ssrA* transcript and consequently read-through into the neighbouring prophage-carrying *speA* gene. Here we report a single SNP in the 5′ transcriptional leader of *ssrA* drives enhanced SpeA superantigen expression in the new M1$_{UK}$ lineage as a result of increased *ssrA* terminator read-through, generating a long bicistronic *ssrA-speA* transcript. Transcriptional read-through has been suggested to occur in approximately one-third of bacterial terminators[50]. In comparison to M1$_{global}$, the molecular mechanism driving enhanced *speA* expression in M1$_{UK}$ is higher levels of transcriptional read-through as a result of the 5′ transcriptional leader *ssrA* SNP increasing complementarity between the 5′ leader of *ssrA* and the T1 terminator.

The emergence of the M1$_{UK}$ lineage in the UK has been epidemiologically linked to increases in invasive disease and seasonal surges of scarlet fever[19,20]. Over the course of this study, neither scarlet fever nor *S. pyogenes* invasive infections were nationally notifiable in Australia. While we have not seen an increase in Queensland notifiable invasive *S. pyogenes*[51] and Queensland Emergency Department Information System scarlet fever numbers in 2020 and 2021, any potential increase may have been mitigated by the public health interventions in response to COVID-19. Comparatively, social distancing measures introduced to combat the COVID-19 pandemic more effectively suppressed other respiratory infections such as pertussis and influenza[51] (Supplementary Table 2). The ongoing replacement of the *S. pyogenes* M1$_{global}$ clone with M1$_{UK}$ in Australia and elsewhere demands heightened vigilance to determine the future clinical impact of this new variant.

## Methods

### Source of Australian *Streptococcus pyogenes* isolates

All 318 Australian *S. pyogenes* isolates were obtained from the Queensland Health Department (Human Research Ethics Committee Reference numbers: HREC/10/QRCH/113 and HEC20-01) or from the Microbiological Diagnostic Unit Public Health Laboratory, Peter Doherty Institute for Infection and Immunity, Melbourne, Victoria (Human Research Ethics Committee Reference number: 1954615) under the Victorian Public Health and Wellbeing Act 2008. These came predominantly from state-based public health reference laboratories in Queensland and Victoria, which together provided 310 invasive isolates from sterile body sites collected between 2005 and 2020. In Queensland, invasive *S. pyogenes* infections are notifiable and 238 invasive isolates originated from this state, while another 72 were from Victoria where such infections became notifiable only in 2022 prior to which, referral to the state public health microbiology laboratory was not a routine requirement. The remaining eight isolates were from the throats of Queensland children with scarlet fever.

### Bacterial strains and growth conditions

*S. pyogenes* strains were grown overnight at 37 °C on 5% horse blood agar and then statically in Todd-Hewitt broth supplemented with 1% yeast extract (THY). Bacteria were routinely inoculated into THY to an optical density at 600 nm (OD$_{600}$) of 0.1 and grown to late-exponential growth phase (OD$_{600}$ of 0.8). *Escherichia coli* strains MC1061 and TOP10 were used for cloning and were grown in Luria–Bertani medium (LB). Where required, spectinomycin was used at 100 μg ml$^{-1}$ (both *S. pyogenes* and *E. coli*). All bacterial strains and plasmids are listed in Supplementary Table 3.

### Illumina genome sequencing

Whole genome sequencing of the clinical isolates was performed by Queensland Health Forensic and Scientific Services ($n = 245$) and Microbiological Diagnostics Laboratory - Public Health Laboratory of Victoria ($n = 72$) Australia using the Illumina NextSeq 500 platform with 150 base pair paired-end chemistry. Reads were trimmed to remove adaptor sequences and low-quality bases with Trimmomatic v0.39 (https://github.com/timflutre/trimmomatic), with kraken used to investigate contamination (v0.10.5-beta, https://github.com/DerrickWood/kraken). Draft genomes were generated using shovill v1.0.9 (https://github.com/tseemann/shovill) with an underlying spades v3.13.0 assembler[52]. Annotation of genes was performed with prokka v1.14.0[53].

### Generation of *S. pyogenes* reference genomes

Genomic DNA of *S. pyogenes* isolates SP1380, SP1384, SP1426, and SP1448 was prepared from solid media scrapings of pure culture using the GenElute Bacterial Genomic DNA Kit (Sigma-Aldrich), and the Gram-positive protocol. High molecular weight DNA was then selected through AMPure-based size selection, using a 0.6× ratio of sample (200 μl) to AMPure XP-beads (120 μl) (Beckman Coulter). Genomic DNA was sequenced in parallel on the Oxford Nanopore Technologies (ONT) GridION and Illumina Nextseq 500.

For ONT sequencing libraries, genomic DNA was prepared according to the manufacturer's protocols using a ligation sequencing kit (ONT), with minor modifications. All mixing steps for DNA samples were done by gently flicking the microfuge tube instead of pipetting and the optional shearing step was omitted. DNA repair treatment was carried out using NEBNext FFPE DNA Repair Mix (New England Biolabs). End repair and A-tailing was performed with NEBNext Ultra II End Repair/dA-tailing Module (New England Biolabs) and sample incubated at 20 °C for 5 min and 65 °C for 5 min. End-repaired products were purified with 1× Agencourt AMPure XP beads. Adapters provided in the respective library kits were ligated to DNA samples with Quick T4 DNA Ligase (New England Biolabs) and samples were incubated at room temperature for 10 min. Purification and loading of adapted libraries on an appropriate flow cell (R9.4.1, ONT) was completed as stated in the manufacturer's protocol and sequenced using the appropriate MinKNOW workflow. The libraries were base called using Guppy v3.0.6.

Reference genomes were assembled using Unicycler v0.4.7 (https://github.com/rrwick/Unicycler) with ONT and Illumina sequence reads from the same DNA preparation and conservative bridging of contigs. Nanopore long read sequences were filtered using filtlong v0.2.0 (https://github.com/rrwick/Filtlong) for the highest quality sequences with selection criteria of >10kb reads and maximum 100× coverage. Final circularized assemblies were annotated using PGAP v4.12 through the National Centre for Biotechnology Information (NCBI). The complete annotated genome assemblies are available at GenBank under the accession numbers CP060267 (SP1448), CP060268 (SP1426), CP060269 (SP1380), and CP060270 (SP1384).

### Comparative genomics

Reference genomes were aligned using MAUVE v2.4.0 genome aligner. Smaller genomic differences were assessed using a custom pipeline based on the tool ekidna v0.3.0 (https://github.com/tseemann/ekidna). In brief, reference genomes were mapped and variants called using paftools as part of minimap2 v2.24[54]. Conserved indels present in all 4 M1$_{UK}$ reference genomes and absent in the 2 M1$_{global}$ reference strains (HKU488, SP1426) were obtained using vcf-isec from VCFtools v0.1.16.

### Population genetics

A database of 736 M1 *S. pyogenes* genomes (317 from this study) and 419 high-quality sequences from publicly available genome sequences across 5 continents was generated (BioProject PRJNA872282, Supplementary Data 1). Illumina paired-end short reads were mapped to the reference sequence (MGAS5005) using BWA-MEM2 as part of snippy v4.6.0 (github/tseemann/snippy) and the core genome alignment determined using snippy-core with default settings. Functional annotations of SNPs and small indels were performed using SnpEff v4.3t[55] as part of snippy and multi-VCF file collated with VCFtools.

The core genome alignment obtained from snippy-core was used for tree building. Regions of irregular SNP density were identified in the MGAS5005 reference genome and the 737 isolate core genome alignment using Gubbins v2.4.0[56]. All low complexity mapping regions, high SNP density regions and known mobile genetic elements were then excised from the alignment resulting in a 1,623,078 bp core genome alignment with a total of 3465 SNP sites consisting of 1,015 parsimony informative and 2450 singleton sites. This consensus SNP alignment was used to build a maximum-likelihood tree with IQ-TREE v1.6.12[57]. A general time-reversible model with gamma correction (GTR + G4) was used, performed with 1000 bootstrap random resamplings to assess tree support. Phylogenetic trees and associated data were visualized using ggtree v2.0.1[58,59], tidyverse v1.3.0[60], phangorn v2.5.5[61], treeio v1.10.0[62] and phytools v0.6-99[63].

## Gene screens and phage comparisons

Virulence factors and genes of interest identified in the mobile genetic elements contained in genome sequences were screened using screen_assembly v1.2.7[64]. Initial screens to detect gene presence were undertaken with 80% identity and 80% length. *emm*-typer commit: 500d048 on branch: master (https://github.com/MDU-PHL/emmtyper) was used to define *S. pyogenes emm* type.

Genetic sequences of prophage from *S. pyogenes* reference genomes were extracted using magphi[65] with seed sequences based on attachment sites described previously[49]. Pairwise sequence alignment of φHKU488.vir and φSP1380.vir (containing *ssa*, *speC*, and *spd*1 virulence genes, located next to *uvrA* insertion site) was determined by tblastN using Easyfig v2.2.2[66].

## Short-read RNA-sequencing and differential gene expression

Total RNA was routinely isolated from bacterial cells using the RNeasy minikit (Qiagen) as previously described[67]. In brief, *S. pyogenes* strains were grown in THY medium to an $OD_{600}$ of -0.8. Two volumes of RNAprotect (Qiagen) were added to the cultures. After 5 min of incubation at room temperature, bacterial cells were collected by centrifugation at $4000 \times g$ for 10 min at 4 °C. RNA was isolated from dry pellets as per the manufacturer's instructions with an additional mechanical lysis step using Lysing Matrix B tubes on the FastPrep-2 5G bead beating grinder and lysis system (MP Biomedicals). To ensure complete removal of contaminating DNA, RNA samples were further purified using the Turbo DNA-free kit (Invitrogen) according to the manufacturer's instructions. RNA-seq analysis was performed at the Australian Centre for Ecogenomics (University of Queensland, Brisbane, Australia). cDNA libraries were prepared from total RNA using TruSeq stranded total RNA library prep with Ribo-Zero Plus rRNA depletion kit (Illumina). Sequencing of the cDNA libraries was performed on the NovaSeq 6000 system (Illumina) on a 2 × 150 bp SP flow cell run generating an average of 20 million reads per sample.

Raw RNA-seq reads were quality assured using FastQC v0.11.0[68] and MultiQC v1.9[69]. TrimGalore v0.6.5 was used to trim Illumina primers (https://github.com/FelixKrueger/TrimGalore). Reads of ribosomal RNA were filtered using SortMeRNA v4.2.0[70] and rRNA extracted from *S. pyogenes* stain SF370, 5448, and HKU488. Reads were aligned to respective reference genomes using BWA-MEM v0.7.17. Reads within features were counted using featureCounts from Subreads v2.0.0[71]. Reads were counted with strand specificity and multi-mapped reads were counted at largest overlapping feature. Differential expression analysis was done using DEseq2 v1.32.0[72] and edgeR v2.23.1[73] in R 4.1.1.

Read coverage plots were constructed using bamCoverage from Deeptools v3.5.0[74], with a bin size of 1, extension of reads, scaling based on all reads, read depth in Counts Per Million reads, and strand specific counting. Bedgraphs were plotted using ggplot2 v3.3.5[75]. The RNA-seq reads and associated gene expression profiles have been deposited in NCBI's Gene Expression Omnibus under the accession number GSE212243.

## Long-read native RNA sequencing

**RNA extraction and poly(A) tailing.** A single colony of SP1380 was inoculated in BHI and incubated at 37 °C overnight. The overnight inoculum was subcultured 1:10 into fresh BHI and cultured to an $OD_{600}$ of -0.8 ± 0.05. The culture was pelleted at 7000 rpm for 2 min, snap frozen on dry ice and stored at −80 °C for subsequent RNA extractions. RNA was extracted as described previously[39] via the PureLink RNA Mini Kit (Thermo Fisher Scientific) in accordance with the manufacturer's protocols, which included using homogenizer columns (Thermo Fisher Scientific). A DNA depletion step was conducted via the TURBO DNA-free kit using 2 U TURBO DNase for 30 min at 37 °C (Thermo Fisher Scientific). DNA-depleted RNA was purified using RNAClean XP beads (1.8× beads: RNA ratio) (Beckman Coulter).

The rRNA was depleted via the MICROBExpress Bacterial mRNA Enrichment Kit (Thermo Fisher Scientific). Minor protocol changes included adding 1 μg of DNA-depleted RNA and the enriched mRNA was precipitated for 3 h at −20 °C. Poly(A) addition was performed using the Poly(A) Polymerase Tailing Kit (Astral Scientific) in accordance with the manufacturer's alternative protocol (4 U input of Poly(A) Polymerase). The input SP1380 RNA concentration was 1 μg, and samples were incubated at 37 °C for 8 min. Poly(A) + RNA was purified using RNAClean XP beads (1.8× beads: RNA ratio) (Beckman Coulter). RNA was quantified using the Qubit RNA HS kit and DNA via the Qubit 1× dsDNA HS kit using a Qubit 4.0 (Thermo Fisher Scientific), purity determined with a NanoDrop 2000 Spectrophotometer (Thermo Fisher Scientific) and size distribution determined via an Agilent RNA ScreenTape on a 4200 TapeStation (Agilent Technologies).

**ONT library preparation and sequencing.** The SP1380 RNA library was prepared using the direct RNA (SQK-RNA002) sequencing kit (input: 450 ng). Sequencing was performed on the ONT MinION platform with R9.4.1 (FLO-MIN106D) flowcells for 72 h and live base-called using Guppy v5.0.17 (High-accuracy model, min_qscore 7). The SP1380 ONT direct RNA reads are available in the NCBI repository BioProject PRJNA872764 (SRR21185202).

**ONT read mapping.** Reads were quality controlled using FastQC v0.11.0[68] and SeqKit v2.2.0 stats[76]. cutadapt v3.8.6[77] was used for filtering small (<75 bp) reads. Reads were aligned to appropriate reference genomes using minimap2 v2.24[54], maximum intron length 100 bp, secondary-to-primary score ratio 0.98, maximum of 2 alignments per transcript, and strand-specific alignment (-u f) for direct-RNA sequencing.

## Determination of *ssrA* relative transcriptional read-through

*ssrA* transcriptional read-through is defined as mean read coverage at genomic regions immediately downstream of proposed *ssrA* transcriptional terminators. Genomic regions are defined per genome: A read-through distribution was determined as mean coverage of genomic regions, normalized to *ssrA* read coverage. Transcriptional read-through was sampled 10,000 times to obtain a relative distribution. Genome coordinates for defining transcriptional read-through were defined as: SF370 *ssrA*, 1,065,025-1,065,372; T1-T2, 1,065,434-1,065,588; post-T2, 1,065,589-1,065,674. Genome coordinates for regions of interest in 5448: *ssrA*, 855,001-855,348; *speA*, 853,686-854,441. Genome coordinates for regions of interest in SP1380: *ssrA*, 1,006,592-1,006,938; post-T1, 1,006,939-1,007,531; *speA*, 1,007,498-1,008,253. Number of samples drawn for bootstrapping equal to base pairs of *ssrA* times the number of biological replicates (n = 3). Refer to Supplementary Fig. 4.

## Construction of isogenic mutants

Isogenic *S. pyogenes* mutants were generated using a highly efficient plasmid (pLZts) for creating markerless isogenic mutants[79]. Briefly, the desired mutation constructs for SP1380$^{ssrA*}$, SP1380$^{rofA*}$, SP1448$^{ssrA*}$ and SP1380$^{rofA*}$ were generated by PCR amplifying the targeted sequence using genomic DNA of *S. pyogenes* M1$_{global}$ strain 5448 as a template. The same protocol was used for the isogenic mutant strain 5448$^{ssrA*}$, using genomic DNA of *S. pyogenes* M1$_{UK}$ strain SP1380 as a template instead. To generate SP1380$^{ssrA-T2}$, ~600 bp of either side of the *speA*-phage integration site was PCR amplified with primer pairs 5′M1$_{UK}$_ssrAT1_F/5′M1$_{UK}$_ssrAT1_R and 3′M1$_{UK}$_ssrAT1_F/3′M1$_{UK}$_ssrAT1_R, using SP1380 as a template. The sequence of the Rho-independent terminator T2 of *ssrA* was PCR amplified with primers M1_ssrAT2_F/M1_ssrAT2_R, using *S. pyogenes* M1 strain SF370 as a template. Point mutations in the T1 terminator stem loop were introduced using the QuikChange II site-directed mutagenesis kit (Agilent). All resulting PCR fragments were cloned into pLZts and used for transformation of competent cells. PCR primer sequences are provided in Supplementary Table 3. Gene deletions were confirmed by DNA sequence analysis (Australian Equine Genome Research Centre, University of Queensland, Brisbane, Australia).

## Quantitative real-time PCR (qPCR)

qPCR was performed using the primers specified in Supplementary Table 3, using SYBR green master mix (Applied Biosystems) according to the manufacturer's instructions. All data were analyzed using QuantStudio Real-Time PCR software v1.1 (QuantStudio 6 Flex, Life Technologies). Relative gene expression was calculated using the threshold cycle ($2^{-\Delta\Delta CT}$) method with *proS* as the reference housekeeping gene[19]. All reactions were performed in triplicate from three independently isolated RNA samples.

## Western blot analyses

*S. pyogenes* strains were routinely grown to late-exponential growth phase in THY. Filter-sterilized culture supernatants were precipitated with 10% trichloroacetic acid (TCA). TCA precipitates were resuspended in loading buffer (normalized to OD$_{600}$). Samples were boiled for 10 min, subjected to SDS-PAGE, and then transferred to polyvinylidene difluoride membranes for detection of immuno-reactive bands using a LI-COR Odyssey Imaging System (LI-COR Biosciences). The primary antibodies used for the detection of SpeA, SpeC, SSA and Spd1 protein in *S. pyogenes* culture supernatants were rabbit antibody to SpeA (PAI111, Toxin Technology; 1:1000 dilution), rabbit antibody to SpeC (PCI333, Toxin Technology; 1:1000 dilution), affinity-purified rabbit antibody to SSA (produced by Mimotopes; 1:500 dilution)[9] and mouse antibody to Spd1 (1:1000 dilution)[10]. Anti-rabbit IgG (H+L) (DyLight 800 4× PEG Conjugate, NEB, 5151P) or anti-mouse IgG (H+L) (DyLight 800 4× PEG Conjugate, NEB, 5257S) were used as the secondary antibodies (1:10,000 dilution).

## Northern blotting

Purified total RNA was quantified using the High-sensitivity (HS) RNA Qubit assay (Thermo). A total of 5 µg of RNA was denatured with fresh glyoxal mixture in a 5:1 ratio for 1 h at 55 °C. Denatured RNA was resolved on a 1% BPTE (100 mM PIPES, 300 mM Bis-Tris, 10 mM EDTA) agarose gel containing SYBR Green (Thermo) and run for 1 h at 100 V in 1× BPTE buffer. SYBR stained ribosomal RNAs were visualized on a Bio-Rad Chemi-doc and used as a loading control. The gel was washed consecutively in 200 mL of 75 mM NaOH, 200 mL of neutralizing solution (1.5 M NaCl and 500 mM Tris-HCl, pH 7.5), and 200 mL of SSC buffer (3 M NaCl and 300 mM sodium citrate, pH 7.0) for 20 min each at room temperature. RNA was capillary transferred onto a Hybond-N+ nylon membrane (GE Healthcare) for 16 h and then UV-crosslinked in a Stratagene Auto-crosslinker with 1200 mJ of UV-C. Pre-hybridization of the membrane was performed using 10 mL of Ambion ULTRAhyb Ultrasensitive hybridization buffer (Thermo) for 30 min at 42 °C. Oligonucleotide probe (5′ – aggaatttctaaatgattccttcatgatttgttaccccctccg –

3′) was radiolabeled with 20 µCi γ32P-ATP (Perkin-Elmer) using T4 polynucleotide kinase (NEB) for 1 h at 37 °C and then purified using a Microspin G-50 column (GE Healthcare). Approximately 10 pmol of γ32P end-labelled probe was incubated with the pre-hybridized membrane for 16 h at 42 °C. The membrane was then washed three times in 2× SSPE (0.3 M NaCl, 20 mM NaH2PO4, 2 mM EDTA) buffer with the addition of 0.1% SDS for 15 min at 42 °C, then imaged using a BAS-IP MS 2040 phosphorscreen on a FLA9500 Typhoon (GE Healthcare).

## Statistical analysis

Differential gene expression from Illumina genome sequence was calculated using DEseq2[72], using a Wald test with Benjamini Hochberg correction for multiple comparison. Batch effects were added in as covariates to the model where indicated. Statistical analysis of qPCR data was performed using Prism software (GraphPad; version 9.4.1). Significance was calculated using one-way analysis of variance (ANOVA) with Dunnett's or Tukey's multiple comparisons post-hoc test or Welch's t-test, where indicated. A *p* value less than 0.05 was determined to be statistically significant.

## Reporting summary

Further information on research design is available in the Nature Portfolio Reporting Summary linked to this article.

## Data availability

The complete annotated genome sequences generated in this study have been deposited in the NCBI database under the BioProject PRJNA656382 with the GenBank accession numbers CP060267 (SP1448), CP060268 (SP1426), CP060269 (SP1380) and CP060270 (SP1384). Illumina short-reads of 318 M1 *S. pyogenes* from Australia have been deposited under the BioProject PRJNA872282. The RNA-seq reads and associated gene expression profiles have been deposited in NCBI's Gene Expression Omnibus under the SuperSeries accession number GSE212243. The SP1380 ONT direct RNA reads are available in the NCBI repository BioProject PRJNA872764 (SRR21185202). Source data are provided with this paper.

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

## Acknowledgements

The work was supported by the National Health and Medical Research Council of Australia. We acknowledge the support of staff at Queensland Health Forensic and Scientific Services (funded by the Queensland Government) and the Microbiological Diagnostic Unit Public Health Laboratory (funded by the Victorian Government). We acknowledge Prof. Kwok-Yung Yuen (Hong Kong University) for providing *S. pyogenes* isolate HKU488.

## Author contributions

M.R.D., N.K., S.B., A.D.I., K.G., A.V.J., and M.J.W. planned the study. M.R.D., N.K., S.B., M.G.J., A.J.C., A.J.H., M.E.P., D.M.P.D.O., N.H.P., O.M.B., D.G.M., B.C., G.T., H.V.S., N.X.F., L.J.M.C., K.S., B.P.H., S.Y.C.T., M.S.S., J.J.T., A.D.I., K.G., A.V.J., and M.J.W. designed experimental procedures, provided reagents and generated data. M.R.D., M.G.J., A.J.H., J.A.L., and A.V.J. managed omic datasets and metadata. M.R.D., N.K., S.B., M.G.J., A.J.H., J.A.L., J.J.T., A.V.J., and M.J.W. analyzed data. M.R.D., N.K., S.B., M.G.J., M.S.S., J.J.T., A.D.I., K.G., A.V.J., and M.J.W. wrote the manuscript. M.R.D., A.V.J., and M.J.W. jointly supervised this work. All authors revised and approved the manuscript.

## Competing interests

The authors declare no competing interests.

## Additional information

[1]Department of Microbiology and Immunology, The University of Melbourne at The Peter Doherty Institute for Infection and Immunity, Melbourne, VIC, Australia. [2]Australian Infectious Diseases Research Centre and School of Chemistry and Molecular Biosciences and Institute for Molecular Bioscience, The University of Queensland, Brisbane, QLD, Australia. [3]School of Biotechnology and Biomolecular Sciences, University of New South Wales, Sydney, NSW, Australia. [4]Department of Infectious Diseases, The University of Melbourne at The Peter Doherty Institute for Infection and Immunity, Melbourne, VIC, Australia. [5]Public Health Microbiology, Queensland Health Forensic and Scientific Services, Queensland Health, Coopers Plains, QLD, Australia. [6]Microbiological Diagnostic Unit Public Health Laboratory, The Department of Microbiology and Immunology, The University of Melbourne at The Peter Doherty Institute for Infection and Immunity, Melbourne, VIC, Australia. [7]Victorian Infectious Diseases Service, The Royal Melbourne Hospital, at the Peter Doherty Institute for Infection and Immunity, Melbourne, VIC, Australia. [8]Illawarra Health and Medical Research Institute and Molecular Horizons, School of Chemistry and Molecular Bioscience, University of Wollongong, Wollongong, NSW, Australia. [9]University of Queensland Centre for Clinical Research, Brisbane, QLD, Australia. [10]Queensland Children's Hospital, Brisbane, QLD, Australia. [11]School of Medicine and Dentistry and Menzies Health Institute Queensland, Griffith University, Gold Coast, QLD, Australia. [12]Departments of Infectious Diseases and Paediatrics, Gold Coast Health, Gold Coast, QLD, Australia. [13]These authors contributed equally: Mark R. Davies, Nadia Keller, Stephan Brouwer, Magnus G. Jespersen.
✉e-mail: mark.davies1@unimelb.edu.au; mark.walker@uq.edu.au

