## [Peer Review File · Nature Communications]

Detection of *Streptococcus pyogenes* M1_{UK} in Australia and characterisation of the mutation driving enhanced expression of superantigen SpeAREVIEWER COMMENTS

Reviewer #1 (Remarks to the Author):

Serotype M1 GAS has proven itself over time to be one of the most impactful and damaging bacterial strains to humans. The rapid emergence of the currently predominant virulent M1 global within a small window of time in the 80s was a dramatic occurrence on the world stage and serves as a warning that this M1-UK might be emerging on meaningful global scale.

This is an impactful report on two fronts. First, it reports the emergence of the M1-UK clone in Australia, which is disconcerting, since GAS strain epidemiology in Australia is well documented to be quite distinct from the western countries where M1-UK has been previously shown to have emerged. Second, it unambiguously describes the mechanism of toxin SpeA over-expression within this subclone (great work!).

I have very few criticisms here in that the work is very well described. What I believe is my only significant suggestion is to better describe features of the surveillance and of the isolates. Are these invasive isolates, non-invasive? Associated with scarlet fever? I could be wrong, but I believe that the description of the majority of these recent M1-uk subclone isolates as invasive GAS "iGAS" is solely described within sTable 4. That to me is a mistake in that this information should be highlighted. In particular, finding impactful emergence within invasives is a demanding assessment of strain success.

Knowing a little more regarding the surveillance population and the the sampling methods would be helpful. On this front, it is my opinion the finding of this subclone in Australian clinical isolates should be mentioned in the title. Even if the surveillance methodology is not optimal, this is a major part of this story to me: "Here we investigate the previously unappreciated expansion of M1UK in Australia, now isolated from the majority of serious infections caused by serotype M1 GAS."

Minor points:

1. Minor point: lines 247-248: Should probably specify that the 36-kb nga-slo event was a replacement event.
2. Lines 250-253: Ref 21 (and perhaps ref 23 as well) is a little misplaced here. It documented the existence of only a small number of the M1-UK lineage (11 of 1052 invasive emm1) recovered during 2015-2018 in the U.S. That said, this sentence is apt in that refs 19, 20, and 22 suggest a startling tendency for this subclone in different surveillance regions to rapidly emerge as a dominant M1 strain.
3. Perhaps it is an over-simplification, but circumstantially it appears that the enhanced expression of speA in M1-UK has been favored by selective pressure and contributes to the fitness as well as the virulence of this strain. I understand that it is speculative to say more, but if you did I would not be critical.

Reviewer #2 (Remarks to the Author):

The authors have characterized the M1uk lineage of *Streptococcus pyogenes* to identify the molecular mechanisms behind the increased expression of SpeA by this lineage. They have found it not to be related to SNPs in the regulator gene *rofA* but instead it is due to an interesting mechanism involving read-through from a neighboring gene. The authors have performed a comprehensive set of experiments that fully support their hypothesis and conclusions.

Minor comments:

Streptococcus pyogenes is use in the manuscript title but occasionally throughout the manuscript but more commonly used is GAS. *Streptococcus pyogenes* is preferred over GAS given that it is not recognized that GAS is not a single species.

Abstract – line 50. The SNP is not 'in the M1uk tmRNA gene'. Can you provide an explanation for 'tmRNA' here too.

Line 70 – UK scarlet-fever strains are not 'classified' as the emm-types listed. These are just the emm-types more commonly associated with scarlet-fever.

Lines 90-91 – how are these isolates sourced? are isolates routinely collected by the reference laboratories? What type of clinical specimens?

Lines 104-115 – this is a little unclear. The authors state 26% have the *ssa/speC/spd1* repertoire and then later say the SP1380 phage is present in 'several' strains. Is this gene repertoire always found on either SP1380 or HKU488? What about just *speC/spd1* (line 127)?

Line 154 – not clear how far upstream *ssrA* is. No comment has been made on paradox gene in between.

Line 167 – Should this be Supplementary Table 5?

Line 187 – at this point you have not shown that the *ssrA* SNP has a critical role in enhanced *speA* expression

Line 227 – the concept of the *ssrA* leader sequence is suddenly introduced and this needs to be clearer. Throughout the manuscript references are made the '*ssrA* SNP' but in fact the SNP is intergenic (as listed in Supplementary Table 3). A better explanation is needed as to where this SNP is in relation to the *ssrA* gene and with regard to the leader sequence.

Line 237 – Are there any possibilities for internal promoters driving the read-through?

How is the expression of the paratox gene effected?

Figure 3B – this is not clear at all. Looks like one read extends beyond the *ssrA* terminator
Supplementary Figure 4. What is being measured is mean read coverage in the *speA* ORF. This is not read-through (as the axis states) from *ssrA* as presumably it includes *speA* transcription alone as well as any read-through reads.

Supplementary Figure 5- Did you use oligonucleotide probes for 23S and 16S rRNA or did you stain the gel with SYBR instead? I suspect the latter given the figure. If not then where are the probe sequences?

The thank the reviewers for their considered and helpful comments. Our point-by-point response is below (blue)

REVIEWER COMMENTS

Reviewer #1 (Remarks to the Author):

I have very few criticisms here in that the work is very well described. What I believe is my only significant suggestion is to better describe features of the surveillance and of the isolates. Are these invasive isolates, non-invasive? Associated with scarlet fever? I could be wrong, but I believe that the description of the majority of these recent M1-uk subclone isolates as invasive GAS “iGAS” is solely described within sTable 4. That to me is a mistake in that this information should be highlighted. In particular, finding impactful emergence within invasives is a demanding assessment of strain success.

RESPONSE: We have added specific details regarding the surveillance components of the study in the methods (new section entitled “*Source of Australian Streptococcus pyogenes isolates*”, line 334-342) and in the results (line 91-95). In Australia, National surveillance for iGAS only began in Feb 2022, and as such, surveillance of iGAS has been limited to voluntary submissions. We were fortunate to collect and sequence all iGAS from state-based collections in Queensland (where invasive isolates have been notifiable since 2016) and available *emm1* iGAS isolates from Victoria.

Line 91-95:

*The emergence of the M1_{UK} lineage in the UK and its detection in other countries^{19,21–23} triggered our investigation of 318 Australian *emm1* *S. pyogenes* isolates. Overall 310/318 were invasive isolates from sterile body sites and sourced from state-based public health laboratories in Queensland and Victoria between 2005 and 2020. The remaining 8 isolates were from the throats of children diagnosed with scarlet fever.*

Line 334-342

Source of Australian Streptococcus pyogenes isolates

*In all, 318 Australian *emm1* *S. pyogenes* isolates were obtained. These came predominantly from state-based public health reference laboratories in Queensland and Victoria, which together provided 310 invasive isolates from sterile body sites collected between 2005 and 2020. In Queensland, invasive *S. pyogenes* infections are notifiable and 238 invasive isolates originated from this state, while another 72 were from Victoria where invasive infections became notifiable only in 2022 and prior to which, referral to the state public health microbiology laboratory was not a routine requirement. The remaining 8 isolates were from the throats of Queensland children with scarlet fever.”*

It is my opinion the finding of this subclone in Australian clinical isolates should be mentioned in the title. Even if the surveillance methodology is not optimal, this is a major part of this story to me: “Here we investigate the previously unappreciated expansion of M1UK in Australia, now isolated from the majority of serious infections caused by serotype M1 GAS.”

RESPONSE: We thank the reviewer for this suggestion. and have appended the title which now reads as follows:

Detection of Streptococcus pyogenes M1_{UK} in Australia and characterisation of the mutation driving enhanced expression of superantigen SpeA

Minor points:

1. Minor point: lines 247-248: Should probably specify that the 36-kb nga-slo event was a replacement event.

RESPONSE: This information has been included in the text (line 256-259).

Line 256-259

Three horizontally acquired genetic events differentiate the M1_{global} clone from other emm1 strains circulating at that time: homologous replacement of a 36-kb chromosomal region encoding the toxins NAD-glycohydrolase and streptolysin O

2. Lines 250-253: Ref 21 (and perhaps ref 23 as well) is a little misplaced here. It documented the existence of only a small number of the M1-UK lineage (11 of 1052 invasive emm1) recovered during 2015-2018 in the U.S. That said, this sentence is apt in that refs 19, 20, and 22 suggest a startling tendency for this subclone in different surveillance regions to rapidly emerge as a dominant M1 strain.

RESPONSE: We have corrected this sentence (lines 262-263) to read as follows:

Lines 262-263

The rapid emergence of the new M1_{UK} variant as the dominant emm1 sub-clone in the UK^{19,20} and the Netherlands²², and subsequent detection in North America^{21,23},

3. Perhaps it is an over-simplification, but circumstantially it appears that the enhanced expression of speA in M1-UK has been favored by selective pressure and contributes to the fitness as well as the virulence of this strain. I understand that it is speculative to say more, but if you did I would not be critical.

RESPONSE: We agree with the inference made by the reviewer. Determination of the subtle, yet clearly altered fitness of M1_{UK} versus M1_{global} requires a systematic evaluation. The evidence from global surveillance clearly provides circumstantial evidence of positive selection, as we have implied in this manuscript (lines 265-271). Future biological and *in vitro* experimentation will likely shed more light on the relative change in fitness of this new sub-clone.

Lines 265-271

We reveal rapid replacement of the M1_{global} genotype with M1_{UK} in cases of severe infections identified in two populous Australian states. Furthermore, 26% of Australian M1_{UK} strains have acquired the bacteriophage-encoded superantigens SSA and SpeC, and the DNase Spd1. This toxin repertoire is over-represented in Asian M1_{global} and M12 isolates causing epidemic scarlet fever^{6,8,9,24}. Bacteriophage mediated horizontal transfer of bacterial

virulence determinants may increase bacterial strain diversity and improve evolutionary fitness^{10,44-47}, driving the expansion of the M1UK lineage in the human population.

Reviewer #2 (Remarks to the Author):

The authors have characterized the M1uk lineage of *Streptococcus pyogenes* to identify the molecular mechanisms behind the increased expression of SpeA by this lineage. They have found it not to be related to SNPs in the regulator gene *rofA* but instead it is due to an interesting mechanism involving read-through from a neighboring gene. The authors have performed a comprehensive set of experiments that fully support their hypothesis and conclusions.

Minor comments:

Streptococcus pyogenes is used in the manuscript title but occasionally throughout the manuscript but more commonly used is GAS. *Streptococcus pyogenes* is preferred over GAS given that it is not recognized that GAS is not a single species.

RESPONSE: We have replaced the use of GAS with *S. pyogenes* throughout the manuscript.

Abstract – line 50. The SNP is not ‘in the M1uk tmRNA gene’. Can you provide an explanation for ‘tmRNA’ here too.

RESPONSE: We have corrected this sentence (line 50-52) as follows:

Lines 50-52

*A single SNP in the 5' transcriptional leader sequence of M1UK transfer-messenger RNA gene *ssrA* drives enhanced SpeA superantigen expression as a result of *ssrA* terminator read-through in the M1UK lineage.*

Line 70 – UK scarlet-fever strains are not ‘classified’ as the emm-types listed. These are just the emm-types more commonly associated with scarlet-fever.

RESPONSE: We have corrected this sentence (line 70-71) as follows:

Lines 70-71

*UK emm-types commonly associated with scarlet fever are *emm1*, *emm12*, *emm3* and *emm4* *S. pyogenes*^{7,14}*

Lines 90-91 – how are these isolates sourced? are isolates routinely collected by the reference laboratories? What type of clinical specimens?

RESPONSE: As indicated in the response to Reviewer 1, we have added more details regarding the source of strains reported in this study to the results and included a new section in the methods relating to the isolate sources.

Lines 104-115 – this is a little unclear. The authors state 26% have the *ssa/speC/spd1* repertoire and then later say the SP1380 phage is present in ‘several’ strains. Is this gene repertoire always found on either SP1380 or HKU488? What about just *speC/spd1* (line

127)?

RESPONSE: “several” referred to the presence of the *ssa/speC/spd1* prophage (Φ SP1380.vir) in the M1_{global} population, not the M1_{UK} population. We have clarified this statement (line 116-119) as follows:

Lines 116-119

Comparative analysis of Φ SP1380.vir with the broader emm1 phage population revealed the presence of Φ SP1380.vir in 5 M1_{global} strains (Figure 1B and 1C), indicative of probable prophage convergence within Australian M1_{global} and M1_{UK} genotypes.

None of the Australian M1_{UK} strains carry the *speC/spd1* prophage which was identified in SP1426 (of the M1_{global} lineage).

Line 154 – not clear how far upstream *ssrA* is. No comment has been made on paratox gene in between.

RESPONSE: The *ssrA* leader sequence SNP is 929bp upstream from the *speA* start codon. The sentence (line 157-160) now reads:

Lines 157-160

*Next, we chose to investigate the SNP (+5G>C) found in the 26 nucleotide transcriptional 5'-leader sequence of tmRNA encoded by the *ssrA* gene²⁹⁻³¹, located ~1kb upstream of the *speA* gene and adjacent to the predicted bacterial attachment site (*attB*) into which *speA*-encoding prophages integrate into the M1_{global} genome³².*

The paratox gene is in the opposite orientation to both *ssrA* and *speA*. It was shown not to be differentially expressed in the RNAseq experiments. We have added a sentence (line 173-177) in the RNAseq section to reflect this observation:

Lines 173-177

*The prophage associated paratox (*ptx*) gene which is located between *ssrA* and *speA* in modern *emm1* genotypes was not differentially transcribed in M1_{UK} compared to M1_{global}, or in the 5' transcriptional leader *ssrA* isogenic mutant set. This finding was to be expected considering that the paratox open reading frame is predicted to be transcribed from the anti-sense strand.*

Line 167 – Should this be Supplementary Table 5?

RESPONSE: Correct, this has been changed (line 140). Thank you for picking this up.

Line 187 – at this point you have not shown that the *ssrA* SNP has a critical role in enhanced *speA* expression

RESPONSE: We used mutational analysis of *ssrA* in both SP1380 and SP1448 M1_{UK} backgrounds to show reversion of the *ssrA* SNP resulted in significant reduction in *speA* transcription and SpeA protein expression (lines 157-165).

Lines 157-165

*Next, we chose to investigate the SNP (+5G>C) found in the 26 nucleotide transcriptional 5'-leader sequence of tmRNA encoded by the *ssrA* gene²⁹⁻³¹, located ~1kb upstream of the *speA* gene and adjacent to the predicted bacterial attachment site (*attB*) into which *speA*-encoding prophages integrate into the *M1_{global}* genome³². The *ssrA* gene encodes a component of the conserved bacterial ribosome rescue system with dual alanine-tRNA-like and mRNA-like properties^{33,34}. Correction of the single 5' transcriptional leader *ssrA* SNP in the *SP1380* and *SP1448 M1_{UK}* genetic backgrounds, to reflect the progenitor *M1_{global}*-like genotype (*SP1380^{ssrA*}*, *SP1448^{ssrA*}*), resulted in a significant reduction in transcripts and protein expression of *SpeA* (Figure 2C and 2D; Supplementary Figure 3).*

Additionally, in the same section we swap the *M1_{UK} ssrA* SNP into the *M1_{global} 5448* background to show enhanced *speA* expression (lines 166-168):

Lines 165-168

*To validate this finding, we introduced the *M1_{UK} ssrA* SNP into the *5448 M1_{global}* genetic background (*5448ssrA**) which resulted in a ~5-fold increase in *speA* transcripts (Figure 2E; Supplementary Figure 3).*

Line 227 – the concept of the *ssrA* leader sequence is suddenly introduced and this needs to be clearer. Throughout the manuscript references are made the ‘*ssrA* SNP’ but in fact the SNP is intergenic (as listed in Supplementary Table 3). A better explanation is needed as to where this SNP is in relation to the *ssrA* gene and with regard to the leader sequence.

RESPONSE: The 5'-leader sequence of the tmRNA had been previously defined (lines 157-160).

Lines 157-160

*Next, we chose to investigate the SNP (+5G>C) found in the 26 nucleotide 5'-leader sequence of the transfer messenger RNA (tmRNA) *ssrA* gene²⁹⁻³¹, located ~1kb upstream of the *speA* gene and adjacent to the predicted bacterial attachment site (*attB*) into which *speA*-encoding prophages integrate into the *M1_{global}* genome³².*

We also acknowledge that some confusion may come from our short-hand use of the ‘*ssrA* SNP’ which technically (as correctly pointed out by the reviewer) is actually a SNP in the 5' transcriptional leader sequence of *ssrA*. The reason why the 5' transcriptional leader sequence is defined as “intergenic” in Supplementary Table 1 is because the 5' leader sequence of *ssrA* does not get corrected annotated by gene annotation software. We therefore refer to this SNP as the “5' transcriptional leader *ssrA* SNP” throughout the revised manuscript.

Line 237 – Are there any possibilities for internal promoters driving the read-through? How is the expression of the *paratox* gene effected?

RESPONSE: As mentioned in the response above, the *paratox* gene is in the opposite orientation to *ssrA* and *speA*. In addition, *paratox* was shown not to be differentially expressed in the RNAseq experiments (Supplementary Table 5). No internal promoter at the

end of *ssrA* is evident, as has been previously reported in the systematic characterisation of *MI_{global}* (strain S119) transcriptional start site study by Glaser and colleagues – reference 38: Rosinski-Chupin, I., Sauvage, E., Fouet, A., Poyart, C. & Glaser, P. Conserved and specific features of *Streptococcus pyogenes* and *Streptococcus agalactiae* transcriptional landscapes. BMC Genomics 20, 236 (2019).

Figure 3B – this is not clear at all. Looks like one read extends beyond the *ssrA* terminator

RESPONSE: All reads represented in the blue box constitute evidence of putative read-through. The variation in transcription length is due to rapid RNA degradation associated with the fact that this analysis was performed using a novel direct long-read RNAseq approach. The fact that the relative read-through number is supported in ‘standard’ Illumina RNAseq of isogenic mutant pairs (Supplementary Figure 4) provides additional support for this hypothesis, as stated in the manuscript.

Lines 188-201

*To investigate how the SNP in the 5' leader of *ssrA* affects downstream *speA* transcription, we analyzed the local read coverage around the *speA*-phage integration site using the SP1380 isogenic strain set (Figure 3A). RNAseq data suggest that 0.25-0.35% of *ssrA* transcripts read past a predicted *ssrA* terminator³⁸ through into the *speA* gene of the Australian *MI_{UK}* SP1380 (Figure 3A, Supplementary Figure 4). This level of *ssrA* transcriptional read-through was equivalent in the SP1380^{rofA*} background, yet 5 times reduced (0.05-0.08 %) in SP1380^{ssrA*} (Figure 3A, Supplementary Figure 4). This change in *ssrA* transcriptional read-through was similar to the increase in *speA* gene transcripts detected by qPCR (Figure 2C). Notably, the transcriptional profile of SP1380^{ssrA*} resembled that of the *MI_{global}* genotype 5448 whereas 5448^{ssrA*} showed enhanced *ssrA* transcriptional read-through (0.23-0.26%), underscoring the critical role of the 5' transcriptional leader *ssrA* SNP in enhanced *speA* expression (Supplementary Figure 4). Transcription of *ssrA* itself remained unchanged in all strains analyzed, indicating that the 5' transcriptional leader *ssrA* SNP does not alter *ssrA* promoter activity in *MI_{UK}*, compared to *MI_{global}* (Figure 3A).*

Supplementary Figure 4. What is being measured is mean read coverage in the *speA* ORF. This is not read-through (as the axis states) from *ssrA* as presumably it includes *speA* transcription alone as well as any read-through reads.

RESPONSE: The reviewer is correct. The Y-axis is reflective of relative transcript abundance at two different regions, even though the amount of read-through from *ssrA* does impact the relative abundance of *speA* transcript. To avoid confusion, we have changed the axis which now reads “Relative transcriptional abundance (percentage)”.

Supplementary Figure 5- Did you use oligonucleotide probes for 23S and 16S rRNA or did you stain the gel with SYBR instead? I suspect the latter given the figure. If not then where are the probe sequences?

RESPONSE: Yes, we used SYBR stained 23S and 16S rRNA as an RNA loading control for the Northern Blot. We have clarified this in the methods section (line 561-562), and have updated the figure legend of Supplementary Figure 5 accordingly.

Lines 561-562

SYBR stained ribosomal RNAs were visualized on a Bio-Rad Chemi-doc and used as a loading control.

Supplementary Figure 5.

SYBR stained 23S and 16S rRNA were used as an RNA loading control.